# Alleviating Molecular-Scale Damages in Silica-Reinforced Natural Rubber Compounds by a Self-Healing Modifier

**DOI:** 10.3390/polym13010039

**Published:** 2020-12-24

**Authors:** Bashir Algaily, Wisut Kaewsakul, Siti Salina Sarkawi, Ekwipoo Kalkornsurapranee

**Affiliations:** 1Polymer Science and Technology, Division of Physical Science, Faculty of Science, Prince of Songkla University, Hat Yai Campus, Songkhla 90110, Thailand; bashir.algaily@gmail.com; 2Department of Physics, Faculty of Science and Technology, Al-Neelain University, Khartoum 11111, Sudan; 3Elastomer Technology and Engineering, Department of Mechanics of Solids, Surfaces and Systems, Faculty of Engineering Technology, University of Twente, P.O. Box 217, 7522 NB Enschede, The Netherlands; 4Malaysian Rubber Board, RRIM Research Station, Sg. Buloh, Selangor 47000, Malaysia; ssalina@lgm.gov.my

**Keywords:** filler, intermolecular reparation, polymer failure, rubber crosslink, composite

## Abstract

The property retentions of silica-reinforced natural rubber vulcanizates with various contents of a self-healing modifier called EMZ, which is based on epoxidized natural rubber (ENR) modified with hydrolyzed maleic anhydride (HMA) as an ester crosslinking agent plus zinc acetate dihydrate (ZAD) as a transesterification catalyst, were investigated. To validate its self-healing efficiency, the molecular-scale damages were introduced to vulcanizates using a tensile stress–strain cyclic test following the Mullins effect concept. The processing characteristics, reinforcing indicators, and physicomechanical and viscoelastic properties of the compounds were evaluated to identify the influences of plausible interactions in the system. Overall results demonstrate that the property retentions are significantly enhanced with increasing EMZ content at elevated treatment temperatures, because the EMZ modifier potentially contributes to reversible linkages leading to the intermolecular reparation of rubber network. Furthermore, a thermally annealing treatment of the damaged vulcanizates at a high temperature, e.g., 120 °C, substantially enhances the property recovery degree, most likely due to an impact of the transesterification reaction of the ester crosslinks adjacent to the molecular damages. This reaction can enable bond interchanges of the ester crosslinks, resulting in the feasibly exchanged positions of the ester crosslinks between the broken rubber molecules and, thus, achievable self-reparation of the damages.

## 1. Introduction

One of the competing features of rubber goods is “durability”. Failures of rubbers originate due to excessive local stresses applied during their operations, leading to matrix breakages starting from a small molecular scale, called micro-damages [1]. Continued or repeated operations of the rubbers will result in the progressive failure of rubber molecular networks growing to a larger failure scale, called macro-damages [2]. This leads to the permanent breakdown of materials, whereby they cannot function to fulfill the desired performance any longer and, thus, need to be disposed of. Earlier disposal or shorter lifespan of a product would contribute to an undesired waste accumulation that in turn requires efficient management to eliminate the wastes [3]. Essential resources, e.g., investment costs, labor, and time, will be consumed to enable the management. In this regard, the improved durability of a product should beneficially reduce the consumption of the aforementioned resources.

A final rubber product has a permanent form stability due to a three-dimensional network chemically tailored to rubber chains. This process is commonly known as vulcanization or curing, in which the rubber molecular chains are crosslinked to one another, giving a macromolecular network structure in the rubber matrix [4]. All conventionally vulcanized rubbers, e.g., tire treads, rubber mounts, and gloves, are classified as thermosetting polymers that cannot be reprocessed or recycled, unlike thermoplastics. The nonrecyclability of vulcanized rubbers is mainly due to this chemical three-dimensional network existent in the materials [5]. There have been attempts to recycle rubber vulcanizates using the devulcanization process. To date, successes toward a certain extent on this devulcanizing technology have been reported but cannot yet be implemented in a full-scale process [6]. A permanent network can be generated through curatives. Conventional crosslinking agents include sulfur and peroxide, and they are the most commonly used for rubber vulcanization [4]; for instance, tires are generally cured with sulfur. Once the permanent rubber network is partially damaged, it is unable to be completely recovered or reformed. Furthermore, the failure of the network will progressively occur as a function of use cycles leading to a continually lowered performance and, therefore, reduced durability of the rubber [7].

To make a rubber product endurable toward operating failures, an initiative to prevent or even auto-internally repair the molecular damages would be one of the promising solutions. External operating parameters are basically constant for each individual application, e.g., a set of tires supports a vehicle under a certain range of dynamic mechanical conditions. While the crosslinks and intermolecular interactions in a rubber vulcanizate are rather complex and variable, micro- and macro-damages can possibly be originated at the main chains, crosslinking bonds, and strong and weak interactions [8]. If the interactions or molecular chains are subject to an excess tension force or degrading conditions [9], these elements will inevitably be torn apart, resulting in molecular-scale damages. Therefore, to allay an ongoing process of network failure, self-reparation of the network via a chemical and/or a physical approach would assist in bringing the network back to an acceptable level. Consequently, this could lead to high-performance retention of the rubbers after operating cycles.

As stated, enabling vulcanized rubber self-healing seems to be a good route for reducing the micro- and macro-damages. Innovation in multifunctional materials has enabled scientists to increase their efforts to develop such elastomeric materials possessing significant self-healing properties with acceptable technical performances [10]. Developing self-healing elastomers is an interesting issue; however, research has still been on a preliminary scale. Once it becomes practical, it is expected to have a high impact on a vast range of applications. For tires, this concept has the potential to prolong their service life and may reduce accidents caused by the explosion of tires while running on the road [7]. Truck tires are basically based on natural rubber (NR) as the major rubber. Previous works recently reported that the introduction of ester crosslinks to epoxidized natural rubber (ENR) could make the final vulcanizates reprocessable due to the unique characteristic of a dynamic network based on ester bonds [11]. This ester crosslink is able to be thermochemically exchanged at elevated temperatures when there is a transesterification catalyst present in the system [12]. In other words, this ester crosslink can alter its bonding position at a high temperature through the transesterification reaction [13]. Prior evidence [14] clearly demonstrated that the rubber pellets of an ester-crosslinked ENR can be remolded giving again a cohesive rubber sheet. This is because the ester bonds at the interface of the vulcanizate pellets can exchange their bonding sites, making chemical crosslinks among rubber chains at the pellet interfaces. It is worth determining if the adaptability of the ester crosslinks enables the recombination of separate molecular networks. Therefore, it would be highly promising if this thermochemically exchangeable ester crosslinking system can contribute to the self-reparation of molecular-scale damages in a rubber matrix after operations.

The present study aims at validating the healing ability of a self-healing modifier based on dicarboxylic acid-modified ENR when used in silica-reinforced natural rubber compounds. The modifier was recently evolved to enable the reprocessability of ENR vulcanizates, as reported elsewhere [15]. The silica NR reference was formulated on the basis of an NR truck tread compound [16]. The self-healing modifier was added to the reference compound with varied loading levels. There were two series of samples depending on how the modifier was added, i.e., extra addition or blending. Mixing behavior and compound properties were monitored. Molecular-scale damages were introduced to vulcanized samples using a tensile stress–strain cyclic test, the commonly known “Mullins effect” [17]. The results of immaculate samples were compared with their counterparts which underwent the first cycle test. The capability for self-reparation of the broken bonds or interactions after molecular-scale damages, as well as the correlation between the viscoelastic properties and property retention of vulcanizates, was the focus of this investigation.

## 2. Materials and Methods

### 2.1. Materials

Natural rubber grade STR5L was used (Chalong Latex Industry Co., Songkhla, Thailand). Compounding ingredients included conventional precipitated silica grade Ultrasil VN3 (Evonik United Silica (Siam) Ltd., Rayong, Thailand), bis-(3-triethoxysilylpropyl) tetrasulfide or TESPT (Si-69 tradename, Evonik, Essen, Germany), white process oil (PTT Public Co. Ltd., Bangkok, Thailand), diphenyl guanidine or DPG, *n*-(1,3-dimethylbutyl)-*n*-phenyl-*p*-phenylenediamine or 6PPD (both from Vessel Chemical Co. Ltd., Bangkok, Thailand), zinc oxide or ZnO (Thai-Lysaght Co. Ltd., Phra Nakhon Si Ayutthaya, Thailand), stearic acid (Imperial Industrial Chemicals Co. Ltd. Bangkok, Thailand), *n*-cyclohexyl-2-benzothiazolesulfenamide or CBS, 2,2,4-trimethyl-1,2-dihydroquinoline or TMQ (both from Lanxess Co. Ltd., Bangkok, Thailand), and sulfur (Siam Chemicals Co. Ltd., Samudprakarn, Thailand). The ingredients for the self-healing modifier based on epoxidized natural rubber (ENR) modified with a dicarboxylic acid and zinc acetate dihydrate included the ENR containing 50 mol.% epoxy groups or ENR-50 (Muang Mai Guthrie PCL, Phuket, Thailand), kaolin clay (Siam Chemicals Co. Ltd., Samudprakarn, Thailand), maleic anhydride or MA (Sigma-Aldrich, Shanghai, China), 1,2-dimethylimidazole or DMI (Alfa Aesar, Kandel, Germany), and zinc acetate dihydrate or ZAD (analytical grade, Ajax Finechem Pty. Ltd., New South Wales, Australia). They were used without further purification.

### 2.2. Sample Preparation

#### 2.2.1. Preparation of the Self-Healing ENR Modifier

ENR-50, HMA (hydrolyzed maleic anhydride), DMI, and ZAD at 100, 3, 5, and 5 phr (parts per hundred of rubber by weight), respectively, were mixed in an internal mixer for 15 min with a rotor speed of 70 rpm at a starting mixer temperature of 40 °C, following a protocol described in a previous work as concisely illustrated in Scheme 1 [14,15]. The obtainable rubber masterbatch was named “EMZ” (epoxidized natural rubber modified with hydrolyzed maleic anhydride plus zinc acetate dihydrate) as an acronym in this context. Regarding the function of the ingredients used, DMI was employed as accelerator for the esterification reaction between the carboxylic acids of HMA and epoxide groups of ENR. It was manually premixed with kaolin clay, an inert filler, prior to introduction into the mixer to facilitate the incorporation of the DMI into the compound. ZAD functions as a transesterification catalyst with efficiency to promote the transesterification reaction at elevated temperatures, shuffling the formed ester bonds and enabling network rearrangement [12,13]. This mechanism allows the crumbled rubber to again reform into a coherent vulcanized sheet [14,15]. Hence, this material is considered a self-healing modifier, as it would potentially promote the intermolecular self-reparation of molecular damages in vulcanizates. The material structure, cure characteristics, and other technical properties of this modifier were recently reported in [14,15].

#### 2.2.2. Compound Preparation

The formulation of silica-reinforced natural rubber compounds used as a rubber matrix is shown in Table 1. Two sets of the compounds were prepared with two different EMZ additions: Set A, extra addition of EMZ at 0 phr, 5 phr, 10 phr, and 15 phr to the compounds coded as the reference, E-05, E-10, and E-15, respectively; Set B, blending of NR/EMZ (i.e., the total amount of rubber was 100 phr) at the ratios of 100/0, 95/5, 90/10, and 85/15, giving the compounds coded as the reference, B-05, B-10, and B15, respectively. The reason for designing these two different series for the present investigation was that the EMZ modifier is based on ENR. Thus, it can be considered as either a compatibilizer or a secondary elastomer; a compatibilizer is commonly added as an extra ingredient to compounds, while rubber is added by blending with the base rubber of compounds.

Mixing of the compounds was performed using an internal mixer (Charoen Tut Co. Ltd., Samut Prakan, Thailand) with a mixing chamber of 300 cm^3^. The mixer was operated at a rotor speed of 60 rpm and a fill factor of 70%. An initial temperature of mixing was adjusted at 100 °C. The dump temperature of all compounds was controlled to be in the range of 140–150 °C, such that an optimum of the hydrophobation or silanization reaction between silica and silane coupling agent was obtainable. To start the mixing process, NR was first masticated on a two-roll mill for 2 min to reduce its viscosity. Then, it was loaded into the internal mixer along with the EMZ modifier and mixed for 2 min. After that, half of the silica and silane coupling agent was added and mixed for 5 min prior to incorporation of the second half of the silica and silane together with white process oil. Other ingredients, i.e., ZnO, stearic acid, TMQ, 6PPD, and half of DPG were added, mixed for 3 min, and finally discharged. The obtainable compound was then sheeted out on a two-roll mill and kept overnight before adding CBS, the other half of the remaining DPG, and sulfur on a two-roll mill. The final compounds were sheeted and kept overnight prior to their property characterizations.

### 2.3. Characterizations

#### 2.3.1. Cure Characteristics and Mooney Viscosity

The cure characteristics of the compounds were determined using a moving die rheometer or MDR (MDR2000, Alpha Technologies, Hudson, OH, USA) with a curing temperature of 150 °C for 30 min. The optimum cure time (T_c90_) and rheometer cure torque were reported and used for press-vulcanizing the compounds into 1.5 mm thick sheets. The investigated compounds were tested for their Mooney viscosity using a Mooney viscometer (MV2000, Alpha Technologies, Alpha Technologies, Ohio, USA) at 100 °C with a large rotor according to ASTM D1646. The value of ML (1 + 4) 100 °C was reported.

#### 2.3.2. Apparent Crosslink Density

Measurements of the apparent crosslink density of vulcanizates were carried out via a swelling method with the vulcanizates in toluene as solvent according to ASTM D471-12a. The cured test specimens of 15 × 15 × 1.5 mm^3^ were immersed in 30 mL of toluene for 7 days at room temperature. The samples were removed from the toluene, and then the weight of the swollen samples was measured. The specimens were dried in a vacuum oven at 75 °C for 24 h, and the dry weight of the specimens was finally measured. The apparent crosslink density (ν) was calculated using the modified Flory–Rehner equation as shown in Equation (1) [18].
(1)ν=−(ln(1−Vr0)+Vr0+χVr02)2Vs(Vr013 − Vr02) 
where *v* is the molar volume of the toluene (i.e., 106.9 cm^3^/mole), *χ* is the Huggins interaction constant (i.e., 0.38 for silica-filled NR) [19], and *V_r0_* is the volume fraction of the swollen rubber, which could be computed using Equations (2) and (3) [20].
(2)$$\frac{V_{r0}}{V_{rf}}=1-m\frac{\theta }{1-\theta }$$
(3)$$m=3c\left(1-V_{r0}^{\frac{1}{3}}\right)+V_{r0}-1$$
where *c* is the parameter of silica–rubber interaction (i.e., 1.17) [19], *θ* is the volume fraction of silica, and *V_rf_* is the volume fraction of rubber in the swollen filled rubber, which can be calculated using Equation (4).
(4)Vrf=w2ρrw2ρr+(w1−w2)ρs,
where *w*_1_ is the weight of the swollen sample, *w*_2_ is the weight of the sample after drying, and $\rho_{s}$ and $\rho_{r}$ are the densities of the solvent used (i.e., toluene, 0.886 g/cm^3^) and of the base rubber (i.e., 0.92 g/cm^3^ for NR), respectively [19].

#### 2.3.3. Bound Rubber Content as Indicative for Filler–Rubber Interaction

Bound rubber content was measured using an uncured sample without curatives. The measurement was carried out under an ammonia atmosphere in order to cleave the physical linkages in the samples, giving only the amount of chemically bound rubber content, i.e., filler–rubber interactions, as well as the lightly crosslinked rubber network [21]. The procedure started with preparing an uncured sample of about 0.2 g. Then, the sample was cut into small pieces and placed in a filter bag made from a metal mesh, i.e., 400 mesh. A bag containing the sample was immersed into 20 mL of toluene at room temperature for 72 h, and the solvent was renewed every day. Afterward, the sample was removed from the toluene and dried for 24 h at 105 °C. Then, the sample was again immersed in 20 mL of toluene at room temperature for 72 h and placed in an ammonia atmosphere; the solvent was renewed every day. Finally, the sample was dried for 24 h in a vacuum hot-air oven at 105 °C. The bound rubber content of each sample was calculated according to the following equation [22]:(5)$$Bound\;rubber\;content\;\left(\%\right)=\;\frac{m_{2}-m_{1}w_{2}}{m_{1}w_{1}},$$
where *m*_1_ is the initial weight of a sample, *m*_2_ is the final weight of the dried sample, *w*_1_ is the rubber fraction in the compound, and *w*_2_ is the silica fraction in the compound.

#### 2.3.4. Tensile Properties

Vulcanizates with a thickness of about 1.5 mm were die-cut into dumbbell-shaped specimens. The tensile tests were carried out using a universal testing machine (Instron 3365, Instron Co. Ltd., Bangkok, Thailand) with a crosshead speed of 500 mm/min according to ASTM D 412 at room temperature. Five specimens per vulcanizate were performed, and the average value was reported.

#### 2.3.5. Payne Effect as Indicative for Filler–Filler Interaction

The Payne effect was measured using a rubber process analyzer (RPA, Alpha Technologies, Ohio, USA). An uncured sample was first vulcanized directly in the RPA at 150 °C for its respective T_c95_ and subsequently cooled to 100 °C prior to the Payne effect test. The test was set with a strain sweep in the range of 0.56–100% at a frequency of 0.50 Hz. The Payne effect was calculated using the data of storage moduli measured, i.e., the difference between the storage shear moduli at 0.56% (G’_0__.56_) and 100% strain (G’_100_).

#### 2.3.6. Mechanical Stress Relaxation Analysis

The stress relaxation test of vulcanizates was carried out using a universal testing machine (Instron 3365, Instron Co. Ltd., Bangkok, Thailand) with a crosshead speed of 100 mm/min. In this experiment, the deformation of a sample was set constant at a specific strain of 70% of the ultimate elongation at break of the sample. This means that each sample was predetermined to know its ultimate tensile properties, particularly elongation at break, prior to conducting the test. The decline in force over 60 min while keeping a constant deformation of the samples was monitored. The decay exponential stress relaxation curves of vulcanizates could be computed for their slope using Equation (6) [23].(6)$$\sigma \left(t\right)=\;\sigma_{0}-kln\left(t\right)$$
where $\sigma_{0}$ is the initial stress subjected to a specimen at the starting stage, and k represents the slope of the line.

#### 2.3.7. Assessing the Self-Reparation of Molecular Damages in Vulcanizates

In general, the “damage” of a material is defined as an undesirable failure occurring in a system that affects its performance; hence, an understanding of the damage nature and the damage progression is essential to suppress the damages, maintaining high retention of the product properties. The damages do not necessarily lead to a suddenly entire loss of material performance [10], but rather a continual deterioration in overall properties, once operated under actual or simulated service conditions. The degree of damage can be evaluated by comparing the two states of a material, i.e., pristine or undamaged versus damaged.

The Mullins effect measurement is considered as an approach for monitoring the damage and recovery degree, since it induces the rupture of physical and chemical bonds [17] inside a material matrix during loading cycles; the rupture immediately progresses after the first loading cycle [24]. This rupture mechanism occurs in all cases, i.e., unfilled and filled or reinforced rubbers; this phenomenon is more pronounced in reinforced rubbers [17]. However, in this study, the damage was observed as a reduction in the material stiffness as a result of microscale defects, e.g., polymer chain and network breakages [25].

By utilizing the Mullins effect concept for the present experiment, the molecular network damages were introduced by applying multiple stress/strain cycles (i.e., 10 cycles) to samples [26]. The vulcanizates with a thickness of about 1.5 mm were die-cut into dumbbell-shaped specimens. A tensile testing machine was used; the initial free length between the clamps was 40 mm. A sample was stretched to its maximum extension ratio λ of 70% relative to the elongation at break of the vulcanizate predetermined with a different sample at a constant crosshead speed of 100 mm/min in ambient conditions. After 10 stretching cycles, the sample was rested at room temperature without pressure for 30 min. Then, the sample was treated for 30 min at different temperatures, i.e., room temperature, as well as 40, 80, and 120 °C, in a vacuum hot-air oven and cooled down to room temperature for 60 min. The stress–strain cyclic test was repeated for the treated samples. The obtained results were compared with that of the pristine counterparts to evaluate the self-reparation degree.

In addition, hysteresis calculated from the area of the first stress–strain cycle was assessed to indicate the change in energy dissipation of vulcanizates [27]. To quantify the recovery degree, the recovery ratio (R) was defined as the value of the hysteresis of a healed sample at a specific temperature (A_healed_) to that of the original (A_pristine_) one, expressed as follows:(7)$$R=\frac{A_{\mathrm{healed}}}{A_{\mathrm{pristine}}}\times 100$$

## 3. Results and Discussion

### 3.1. Relationship among Mixing Data, Compound Viscosity, and Filler Interactions

The mixing fingerprints of the investigated compounds are shown in Figure 1. The final mixing torques, i.e., from about 700–900 s of mixing time, more or less leveling off (see Figure 1a). These data technically confirms that the particles of additives, in particular silica filler, are incorporated and broken down to their smallest sizes that cannot be reduced any further under this mixing condition. Thus, the mixings were considered to be completed, as the optimum dispersion degree of the particulate compositions is achievable. In general, the mixing is the most critical processing step determining the quality of final compounds and vulcanizates. These mixing data show that all compounds reach an optimum level in terms of the dispersion degree. A noticeable difference between these compounds is the mixing torque intensity, especially at the final mixing stage.

The dump or discharge temperatures (Figure 1b) and mixing torques of the compounds could be extracted and plotted as a function of EMZ loading with the two different addition procedures, as depicted in Figure 2. Upon increasing the content of EMZ, the dump temperatures of the silica-filled NR compounds increased, reflecting that the mixes generated a higher shear rate that in turn led to a higher thermal accumulation in the compounds. Considering the factor(s) causing this consequence, the dump temperatures corresponded well with the mixing torques (Figure 2) as a result of the increase in compound viscosities during mixing. Therefore, during mixing, these compounds must generate different degrees of intermolecular networking reactions and/or interactions among reactive sites, which contribute to the higher viscosities. In addition, the EMZ can act as a coupling agent which shields the silica surface and better interacts with NR chains, contributing to an increase in the Mooney viscosity, as discussed later for the bound rubber results. The mixing torques, dump temperatures, and Mooney viscosities showed relatively good correlations, as illustrated in Figure 3.

As mentioned above, important intermolecular reactions/interactions must have occurred during mixing, which led to an increase in compound viscosities, as indicated by the mixing torques and Mooney viscosities. This resulted in increased mixing temperatures (see Figure 3). Inside these compounds, there were several possible simultaneous interactions due to the reactive ingredients and a rather high mixing temperature, i.e., 130–150 °C. These ingredients included natural rubber, silica, TESPT silane, accelerators, activators, and the EMZ modifier. The investigated compounds had different quantities of the EMZ masterbatch, while the dosages of other compositions remained identical. Hence, this EMZ component could be the key influencing factor.

Regarding the reference compound which was based on a silica–silane-filled NR system, silica can react with bifunctional organosilane, i.e., bis-triethoxysilylpropyl tetrasulfide or TESPT, under temperatures as low as 120 °C, known as the “silanization reaction”. This reaction becomes more pronounced under elevated temperatures. The silanization of the silane hydrophobizes the polar surface of silica, resulting in less polarity, which provides better compatibility with the nonpolar rubber matrix, i.e., NR in this case. Thus, the dispersion of silica in NR is much better if compared with a system without a silane coupling agent [28]. The amelioration in filler dispersion should reduce the viscosity of the compounds, as discussed later for the results regarding the filler–filler interaction or Payne effect. However, when the TESPT molecule couples with silica clusters, there is a possibility that its molecule splits off into two halves, creating reactive sulfur radicals at the end half-molecule. This site is highly reactive toward NR (diene rubber) molecules and can form a covalent bridge between silica and rubber during elevated temperatures, e.g., this mixing condition [16], leading to a so-called higher “filler–rubber interaction”. This interaction contributes to a higher viscosity of the compounds. In addition, TESPT has a side effect of releasing free sulfur into the system during mixing at high temperature, since it has four sulfur atoms on average in the middle of its molecule. When it snaps during mixing, it can be assumed that each half of the silane molecule carries one sulfur atom, meaning that there are two atoms of sulfur on average for each TESPT molecule liberated into the system. This free sulfur can partially crosslink NR molecules during mixing, contributing to the polymer network and, thus, increasing the compound viscosity [16]. All investigated compounds contained the same dosages of silica, silane, and curative. Therefore, the effects of silica–silane–rubber coupling and lightly crosslinked rubber were assumed to be identical among these compounds. These unique phenomena of the silica–TESPT-filled NR compound further revealed that the reactive sites originating from silica and silane were most likely to interact with the self-healing EMZ modifier.

The self-healing EMZ modifier contains hydroxyl groups from the opened structure of epoxide or oxirane rings on ENR molecules [15]. A direct condensation reaction between the silanol groups on the silica surface and the radical ions at the opened epoxide positions on ENR molecules [21] could occur during mixing due to their acidic character and thermal acceleration. This led to linkage formation between EMZ and silica. The increased degree of this EMZ–silica reaction induced the filler–rubber interactions in the system, as clearly indicated by the bound rubber contents shown in Figure 4, resulting in a higher viscosity and, thus, elevated shear rates, which in turn contributed to a higher mixing temperature (Figure 2 and Figure 3). Therefore, these results essentially hint that the chemical interactions between EMZ modifier and silica and/or the self-association of the EMZ itself, i.e., the reactions between HMA and zinc acetate with ENR-50, can be activated by an effective high temperature during mixing, i.e., 135–145 °C.

The increase in compound viscosities with a higher loading of EMZ, as shown in Figure 3, can be attributed not only to the enhancement in silica–rubber interactions, but also other parameters contributing toward increased compound viscosity. In general, a better silica dispersion resulting from a lower filler–filler interaction plays a key role in reducing the viscosity of the compounds [29]. In this study, TESPT was used at its optimum level (i.e., silica/TESPT: 55/5 phr) [16]; this TESPT content was basically sufficient to hydrophobize the silica surface, giving reduced compound viscosities. However, the free sulfur released from TESPT during mixing could partially crosslink the rubber chains, leading to increased compound viscosities [16]. Moreover, the addition of EMZ could contribute to higher compound viscosities due to reactive interactions. There were two possible reactions occurring during mixing responsible for the rise in compound viscosity upon increasing the EMZ level: (1) The self-association of EMZ through hydrogen bonding or polar interactions between HMA and ENR-50 or even the possible self-crosslinking of ENR [30,31], and (2) chemical interactions generated between silica and ENR [32,33]. These two intermolecular interactions seem to involve in the formation of extra networks, contributing to the increase in viscosities of silica-filled NR compounds.

The filler–filler interactions of the compounds monitored using a Payne effect analysis were evaluated, as shown in Figure 4. The Payne effect of silica-filled NR compounds slightly decreased with increasing EMZ content. This result implies that the interfacial interactions between the silica and rubber matrix were enhanced after the addition of EMZ, giving a greatly reduced filler–filler interaction. As mentioned above, the intermolecular interactions originated through strong covalent and/or hydrogen bonds between the ENR and silica. This also corresponded well with the improvement in bound rubber contents (Figure 4), indicating a better silica–rubber interaction. Therefore, the formation of strong interactions between the silica and ENR-50 consequently accounted for a better silica dispersion in the compound with increasing EMZ loading.

### 3.2. Cure Characteristics and Apparent Crosslink Density

The cure characteristics of silica-filled NR compounds, i.e., scorch time (T_s2_), optimum cure time (T_c90_), minimum torque (S’_Min_), and maximum torque (S’_Max_), are shown in Table 2. With increasing EMZ loading, the scorch and cure times considerably decreased. It was previously reported that the silica surfaces are reactive toward curative absorption, particularly accelerators and ZnO, since these chemicals are polar, which in turn results in cure retardation. Hence, the interactions between EMZ and the silanol groups of silica led to reduced absorption of the curatives on the silica surface and, thus, a faster cure rate of the compounds [34] upon increasing EMZ loading. Additionally, the ENRs contained epoxide groups along their chains that could activate the adjacent double bonds to be more reactive toward creating a radical at an allylic position on the rubber chains. Thus, the epoxide group could also increase the vulcanization rate and reduce the scorch and cure times [35]. The compounds of Set B possessed a higher quantity of EMZ or ENR compared to their counterparts of Set A. Therefore, the cure and scorch times of Set B compounds were slightly faster than those of Set A compounds.

The rheometer minimum torques slightly increased with increasing EMZ content, which corresponded to the increasing trends of Mooney viscosities and the final mixing torque (see Figure 3). The rise in the maximum torque values with increasing content of EMZ supported the higher overall network densities in the system, as indicated by the results of bound rubber contents (Figure 4) and apparent crosslink densities (Figure 5). For filled compounds, the cure torque difference was affected not only by the crosslink density of the vulcanizate but also the filler–filler interaction. However, according to the Payne effect shown in Figure 4, there was a clear decrease in filler–filler interaction. Thus, it is obvious that the network densities of Set B vulcanizates were greater than those of their Set A counterparts. The self-healing EMZ modifier showed its unique characteristic of increasing the network density, i.e., polymer crosslinks and filler–rubber interactions, in this silica-filled NR system.

### 3.3. Reinforcing, Mechanical, and Viscoelastic Properties

The reinforcement index, i.e., M300/M100 [29], is commonly used to indicate the reinforcing efficiency of reinforced rubber compounds, especially in tire technology. In general, this reinforcement index corresponds well with tan δ at 60 °C derived from a dynamic mechanical analysis (DMA), whereby a higher M300/M100 or lower tan δ at 60 °C denotes a higher reinforcing efficiency, which theoretically leads to the better rolling resistance of a tire [36]. According to Figure 6, the reinforcement indices of silica-filled NR vulcanizates showed an increasing trend with increasing EMZ loading. This result is not surprising, since more filler–rubber interactions were generated by EMZ (as discussed in Figure 6) in this system, giving better homogeneity of the mix and stronger linkages between silica and rubber, consequently leading to enhanced reinforcement power.

Figure 6 shows the tensile strength and elongation at break of silica-filled NR vulcanizates containing various EMZ contents. It is obvious that, upon increasing the concentration of EMZ, the elongation at break slightly decreased for the vulcanizates with EMZ from 5 to 10 phr and significantly decreased for the vulcanizates with EMZ at 15 phr (approximately 50 units different), attributed to an increase in apparent crosslink density. The improvement in tensile strength with increasing content of EMZ was consistent with the improved reinforcement index and could also be attributed to a higher filler–rubber interaction, as indicated by the bound rubber content and apparent crosslink density (see Figure 4 and Figure 5).

Figure 7 displays the decay of tensile stress as a function of time for the vulcanizates, characterized by a stress relaxation test. The slope of each stress relaxation line indicates the viscoelastic properties, i.e., the elastic and viscous ratio, of a vulcanized elastomer. The stress relaxation is associated with the continually progressive breakage of networks, interactions, and/or molecular chains, as well as the movement of viscous segments, e.g., small molecules and uncrosslinked short chains, under applied deformation [37]. The slopes of the vulcanized composites with a higher EMZ content were slightly steeper than those of the samples without or with a lower loading of EMZ. This reflects that the EMZ promoted a greater stress relaxation rate of vulcanizates, implying that an increment in EMZ content in the vulcanizates contributed to a higher viscous proportion, as well as to a potentially increased extent of intermolecular interactions and/or short chains [38,39]. A detailed discussion on the involved interactions in this system is provided in Section 3.5. Furthermore, the molecular weight of NR could be reduced due to the EMZ modifier. It is commonly known that hydrolyzed maleic acid or dicarboxylic acid, used as an ester crosslinking agent, can shorten the diene rubber chains due to their acidic characteristics. This leads to a greater extent of shorter molecular chains, a lower average molecular weight, or a more viscous segment, thereby increasing the stress relaxation rate of vulcanizates, as indicated by a higher absolute value of the stress relaxation slope.

Considering the initial stress of each sample, it increased with increasing EMZ content (see Figure 7). This was attributed to the extra crosslinks generated by the self-healing modifier EMZ (see Figure 4 and Figure 5). These additional crosslinks were formed through ester, ionic, or hydrogen bonds. The functionalities of EMZ are also beneficial for enhancing silica–rubber interactions and, thus, improving reinforcing efficiency [40]; more details are discussed in Section 3.5.

### 3.4. Property Retention after Molecular Damages

The stress–strain cyclic test, often known as the Mullins effect, was performed to verify the reversibility of molecular interactions, e.g., ionic and hydrogen bonds, intermolecular forces, and loosely physical entanglements, after introducing molecular damages through a deformation process to a vulcanizate. Samples were stretched to a certain strain of 70% of their predetermined ultimate elongation at break. At this applied strain, some bonds, molecules, or intermolecular linkages were likely ruptured, leading to micro-damages inside the vulcanizate samples. The possible micro-damages could have occurred at filler–filler interactions, loose rubber–filler interactions, short rubber molecules between crosslinking points, and/or entanglements [24]. As evidenced in previous work [17], an obviously sharp decrease in the instantaneous tensile stress of the material in the second test cycle was reported. This is a common phenomenon of a vulcanizate. It can be seen that the second cycle had a much smaller hysteresis (see Figure 8a–c). This is because the major breakages of molecules/intermolecular interactions occurred in the first stress–strain cycle. However, the values derived from the following cycle could imply some immediate reversible interactions in the system. The hysteresis of vulcanizates can be characterized by the difference between the integral areas of the loading and unloading stress–strain curves [17] (see Figure 8d). An obtainable value indicates the energy dissipation from the rupture of bonds/interactions under a large deformation.

As a function of the hysteresis values derived from the integral areas under the cyclic stress–strain curves of samples shown as examples in Figure 9, the degree of hysteresis recovery after micro-damage could be calculated. Figure 10 shows the resulting recovery degree of the investigated samples. Upon increasing the amount of the EMZ modifier, together with elevating the annealing temperature after a micro-damage, the property retention was substantially enhanced. A higher recovery (about 84%) compared to the reference (Ref.) sample was shown for the B-15 sample after undergoing a thermal treatment at 120 °C. This result essentially suggests that there were plausibly reversible elements in the systems. On the other hand, for the reference sample, the recovery ratio showed a significantly smaller value due to a lesser extent of reversible segments in the vulcanizate. Thus, it is obvious that the micro-damage could be alleviated by adding the self-healing modifier EMZ. In addition, elevated temperatures could significantly accelerate the recovery performance. This is common for a chemical reaction/interaction, as higher temperatures raise the reaction kinetics. Moreover, this system of silica-filled NR plus ENR-based self-healing modifier contained rather complex interactions, since there were various reactive sites which could link to one another, particularly through chemical mechanisms.

Regarding the similar trend of recovery ratio observed for the reference sample, it is commonly known that, in a silica-filled rubber compound, there are always reversible interactions in the system. These interactions by nature originate from the reactive ingredients formulated in the compound such as active fillers (silica), as well as from polar species in the system, e.g., protein in natural rubber molecules and the like. Moreover, dis- and re-entanglement of the rubber chains can also play a role in the property recovery [41]. Further information on the possible intermolecular interactions is provided in the next section.

### 3.5. Relevant Intermolecular Interactions in this System

The possible intermolecular interactions could be summarized into a graphical scheme, as illustrated in Scheme 2. These interactions could be categorized into two main types, permanent and reversible linkages, as described below.

Permanent interactions.—These include rubber–rubber crosslinks derived from monosulfidic (C–S–C), disulfidic (C–S_2_–C), polysulfidic (C–S_x_–C), carbon–carbon (C–C), ether, and ester linkages. The sulfidic crosslinks were the majority in this case since the compounds were vulcanized with an accelerated sulfur cure system. In addition, free sulfur liberated from the silane coupling agent “TESPT” could crosslink the rubber molecules, contributing to the network density [42]. For the ENR-based self-healing modifier, direct silica–epoxide coupling could have occurred, forming strong interactions between ENR molecules and silica particles in addition to the coupling through the silane bridging mechanism [32,43]. The network derived from ENR-based self-healing modifier was considered as the minority in this system; ether crosslinks (C–O–C) can be formed through the self-crosslinking of hydroxyl groups from the opened epoxide structure of ENR [30,43,44], and ester linkages can also be generated from carboxylic groups of the hydrolyzed maleic anhydride [14,15]. Moreover, tightly entangled rubber chains are also considered permanent crosslinks. These permanent interactions are harder to be damaged by applied forces compared to ionic, hydrogen bond, polar, and other weak chemical and physical interactions.

Reversible interactions—The compounds prepared in this study were combined with an ENR-based self-healing modifier, i.e., EMZ. This modifier likely contributed a great extent to the reversible linkages to the rubber network. As illustrated in Scheme 2, the possible reversible interactions include hydrogen bonding, as well as rubber–rubber and rubber–filler polar interactions. As described in Scheme 1, the EMZ modifier has the ability to generate ester crosslinks between ENR molecules with the opened oxirane structure. Despite the ester bonds, ionic interactions could have occurred through Zn^2^^+^ and carboxylic groups from the dicarboxylic acids modifying the ENR molecules [12,14,15]. Moreover, ENR and silica can form reversible hydrogen bonding between the silanol groups of silica and the epoxide groups of ENR [21,32,33,34,43]. These interactions could be reversed after their breakage. Thermal treatment could accelerate the reversible mechanism of these interactions. Silica clusters have a strong affinity among themselves. They tend to form larger clusters via hydrogen bonding due to a high concentration of silanol groups on the silica surface. Large silica clusters/network are easy to break down under an applied force [28]. However, they can be recovered when a vulcanizate returns to its original shape. These reversible interactions are considered to be reformed to a certain extent after micro-damages under an unloading condition. In this system, it is most likely that the reversible interactions played a major role in the property retention of the samples, particularly with a higher amount of the healing modifier and an elevated temperature. Moreover, van der Waals forces are additional interactions that can cause the reversibility of a vulcanizate. Large or small molecules possessing similar chemistry have a good affinity between each other, which leads to their good compatibility; these interactions are relatively weak but could confer materials with coherent characteristics.

Physical chain entanglements—In a rubber network, there are tightly and loosely entangled chains. Tight chain entanglements are hardly loosened and are, thus, considered as similar to chemical crosslinks. A high extension results in the removal of loose entanglements, thereby lowering the stress on the second and subsequent loading–unloading cycles, as evidenced in Figure 8a. This would result in a change in material entropy [45]. However, it could partially be recovered by a high temperature. The recovery is supported by thermal motions which produce new physical chain entanglements [46]. It was remarked that thermal treatment can only accelerate the return of rubber chains to their equilibrium state but cannot assist in recovering the ruptured irreversible bonds [47]; this type of bond breakage is regarded as permanent molecular damage.

However, the calculated apparent recovery ratio clearly confirmed good performance after three experimental repetitions. The recovery ratios slightly decreased after the first repetition of the experiment, as shown in Figure 11. In this study, the recovery process was dominated by the reversibility of crosslinked networks in the system, permitting the crosslinked network to be rearranged during thermal treatment. Upon heating, the fractured portions of the network diffused and re-entangled between the molecular chains, causing the ruptured crosslinks to restore their integrity, most possibly due to the exchangeable transesterification interaction and the thermo-reversible hydrogen and/or ionic bonds. As reported in our previous work [15], the transesterification reaction of ester crosslinks/bonds can be accelerated by thermal treatment, whereby a higher treatment temperature (e.g., from room temperature to 200 °C) results in a higher bond-interchange degree, leading to a greater potential for intermolecular self-reparation of the rubber network. In addition, this bond-exchange reaction requires a transesterification catalyst, which was ZAD in this case, whereas the epoxide groups on ENR molecules also needed to be opened, providing radical ions and OH groups to the system, which could react with the carboxylic groups in the self-healing modifier. A chemical reaction regarding this bond-interchange mechanism was proposed in previous works [12,13,15].

### 3.6. Potential Role of the Thermochemically Exchangeable Ester Bonds

In addition to the self-association of the aforementioned reversible interactions that highly contributed to the intermolecular self-reparation of the molecular damages in the silica-filled NR vulcanizates, it is highly promising that the thermochemically exchangeable ester bond from the EMZ self-healing modifier significantly promoted this self-reparation. The possible ester crosslink structure of the modifier is described in Scheme 1. This is because, when the sample was treated at elevated temperatures, particularly at 120 °C, the hysteresis retention showed its highest value (Figure 9), meaning that the elasticity of the material was highly recovered. The reasons for the highly maintained elasticity of a vulcanizate are the rubber network density and a low extent of molecular damages. Treatment of the vulcanizates at 120 °C seemed to be crucial for the self-reparation of the molecular damages through a transesterification reaction of the ester crosslinks. It is most likely that, at the interfaces of the molecular damages, which could have been in the form of micro-voids, the ester crosslinks at one interface of the void could rearrange and exchange the bonding positions to another side of the interface. This generated new ester crosslinks bridging the ruptured interfaces of the void, leading to the micro-damages being repaired. This elucidation is consistent with the findings from previous work [12,14,15]. Scheme 3 graphically illustrates the mechanism of self-reparation of molecular damages through a transesterification reaction of the thermochemically exchangeable ester crosslinks.

## 4. Conclusions

The mixing torques, mixing temperatures, and compound viscosities indicated that adding EMZ contributes to higher intermolecular interactions inside the compounds. Extra polymer networks and/or silica–rubber interactions with the presence of EMZ were confirmed by the results of chemically bound rubber contents and apparent crosslink densities. Furthermore, the EMZ could reduce the filler–filler interactions of silica clusters, as indicated by the lower Payne effect values. The addition of EMZ via blending with the primary rubber had a slightly stronger impact on these properties than that via its extra addition to the compounds, attributed to a higher concentration of EMZ relative to the total rubber content for the blending approach. The strength properties and reinforcement efficiencies of vulcanizates were enhanced with increasing EMZ content. The property retention of vulcanizates was significantly improved with increasing EMZ content and with a thermally annealing treatment at high temperatures, e.g., 120 °C, used as the maximum temperature in this study. This was attributed to the EMZ contributing to ester crosslinks and an increased concentration of chemically reversible linkages. The ester bonds could have potentially been interchanged via a transesterification reaction, which enabled the rearrangement of chemical ester crosslinks in the rubber network, thereby leading to self-reparation of the damaged network. This work clearly confirms that the EMZ self-healing modifier significantly improved the property retention of vulcanizates. The intermolecular self-healing mechanism of this modifier requires a thermal treatment and a transesterification catalyst after unloading conditions. Nonetheless, there are still a number of open questions that remain to be explored in order to explicitly elucidate the effects and mechanisms behind these findings. Further proof of the self-reparation of the molecular damages using advanced characterizations is highly desired.

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
