# Peer review of "Alleviating Molecular-Scale Damages in Silica-Reinforced Natural Rubber Compounds by a Self-Healing Modifier"

_polymers, 2020, doi:10.3390/polym13010039_

Round 1
Reviewer 1 Report
This paper studied the healing ability of a self-assembling modifier based on dicarboxylic acid-modified ENR when used in silica-reinforced natural rubber compounds. The self-healing ability of broken bonds or interactions after molecular scale damage and the correlation between viscoelastic properties and performance retention of vulcanizates were studied. The background and significance of the research are very wonderful, and the research content is also very detailed, the whole research is complete and innovative. I think the current form this work is already for publication. But some minor errors need to be corrected before publication.
- The figures in the manuscript are not beautiful and needs to be revised.
- Line 517-561. If the author needs to make a clause statement, use the normal format.
- The format of references needs carefully revised.
Author Response
Dear highly learned reviewer,
We are highly thankful for your feedback given to this manuscript. We have modified the manuscript following your suggestions and comments as can be seen in the attached file.
Best regards,
Wisut Kaewsakul
Correspondence of the manuscript

Reviewer 2 Report
After I have finished reading the manuscript from the beginning to the end, I cannot find serious weaknesses of this manuscript. I have a few suggestions about the structures and reinforcement mechanism of the silica-reinforced ENR nanocomposites. First, the revised manuscript should include the data and explanations for the structural determination of materials. Thus, the structures of the resulting polymers and nanocomposites can be further analyzed in detail by X-ray photoelectron spectroscopy, solid-state nuclear magnetic resonance spectroscopy and X-ray diffraction. Second, according to Figure 11, what kind of noncovalent complex system with specific functionality did the author employ in this study? Please ensure that the discussion of the complex mechanism of silica/ENR is present in the main text. Is it possible to spontaneously form phase-separated structures with microscopic length scales due to the presence of inorganic material inside cross-linked ENR matrix? Third, information on the experimental evidences of the transesterification reaction is missing from the main text. Finally, as mentioned in the topic: “Alleviating Molecular-Scale Damages in Silica-Reinforced Natural Rubber Compounds by a Self-Assembling Modifier.”, however, self-assembly is a physical process where complex and functional structures are created in a bottom-up manner by the organization of a large set of components.. Thus, I strongly recommend that authors can change the "Self-Assembly" subject by introducing an appropriate word.
Based on the above suggestions, I recommend this manuscript to be accepted after appropriate revision.
Author Response

(The authors gave the same response as above.)

Round 2
Reviewer 2 Report
In reviewing the revised manuscript and point-by-point response to reviewers' comments, the authors have addressed my concerns and thus this paper would be suitable for publication in Polymers.